

# Chemical analysis and angiotensin I-converting enzyme inhibitory activity of enzymatic hydrolysates derived from meat of goat-kids with supplemental selenium

Silvia C. Pérez-Ramirez[1,*], Rosy Cruz-Monterrosa[2,*], Mayra Diaz-Ramirez[2], Erika B. León-Espinosa[3], José E. Aguilar-Toalá[2], Monzerrat Rosas-Espejel[2] and J. Efren Ramirez-Bribiesca[1]

[1] Animal Science, Colegio de Postgraduados, Texcoco, Edo. Mexico, Mexico
[2] Ciencias de los Alimentos, Universidad Autonoma Metropolitana Lerma, Lerma de Villada, Edo. Mexico, Mexico
[3] Tecnologico Nacional de Mexico, Tecnologico de Estudios Superiores de San Felipe del Progreso, San Felipe Progreso, Edo. Mexico, Mexico

[*] These authors contributed equally to this work.

Corresponding author
J. Efren Ramirez-Bribiesca,
efrenrb@colpos.mx

## ABSTRACT

**Background**. The effects of selenium (Se) on animal health due to its antioxidant and immune system regulatory properties are very well-documented. However, there is still a lack of scientific evidence about the effect of Se on muscle tissue. Se supplementation in ruminants will enhance the antioxidant activity of myocytes and increase angiotensin-converting enzyme (ACE) inhibitory activity. Generating bioactive peptides derived from meat could prevent the production of angiotensin II, a key player in the development of cardiovascular diseases.

**Methods**. Forty-five suckling goat kids were randomized into one of three groups: (1) CG: group without Se supplemented in the diet; (2) GSS: group with a single injectable dose subcutaneously of sodium selenite ($Na_2SeO_3$) at a dose of 0.25 mg/kg of body weight; (3) GSM: group with an oral administration of selenomethionine (SeMe) at a dose of 0.3 mg/kg of body weight). The effect of both sources of Se was evaluated on the proximate composition of meat and liver and the angiotensin I-converting enzyme inhibitory activity of meat-derived enzymatic hydrolysates.

**Results**. The kids-goat meat from the GSM group had a higher protein content ($p < 0.05$). The fat content gradually increased over time in the treatment GSM, which increased *ca.* two-fold (from 1.77 to 3.68). The degree of hydrolysis of the meat samples decreased ($p < 0.05$) in the treatments supplemented with Se (GSS and GSM). The degree of hydrolysis increased significantly ($p < 0.05$) over time in treatments with Se (GSS and GSM). The electrophoretic patterns of the enzymatic hydrolysates at two h showed a molecular weight between 23.44 and 27.5 kDa, the bands with more intensity. At 21 d of slaughter, a major degree of hydrolysis was observed in the treatments supplemented with Se (GSS and GSM) compared to the CG. Meat protein content and rate of ACE inhibition after hydrolysis improved (50% and 2%, $p < 0.05$) with GSM at 7 d of slaughter. After hydrolysis, the IC50 of the selenium-supplemented groups decreased ($p < 0.05$) the amount of CAE and IC50 values.

**Conclusion.** This is the first report describing the ACE inhibitory activity of bioactive peptides derived from goat-kids meat with supplemental. These results indicate the presence of ACE in goat meat; however, the percentage of ACE inhibition after hydrolysis was only improved with selenomethionine dosing at 7 days of slaughter. The study's novelty indicates that supplemented selenium synergized with ACE in goat meat. It is necessary to continue these studies to identify specific bioactive peptides, antioxidant activities, and goat meat's biological and functional value, considering it a functional food that can prevent metabolic diseases and be a healthy alternative for the human population.

# INTRODUCTION

Cardiovascular diseases associated with angiotensin II occur due to the activation of nicotinamide adenine dinucleotide phosphate (NADPH) oxidases, mitochondrial dysfunction, inflammation, and the reduction of endogenous antioxidant enzymes (*Koju et al., 2019*). Nitric oxide synthase (NOS) is another source of $O_2$ in which the substrate arginine or the cofactor tetrahydrobiopterin is limited. The uncoupling of NOS occurs due to pathological conditions associated with angiotensin II, such as hypertension, atherosclerosis, and diabetes (*Loperena & Harrison, 2016*). Sources of $O_2$ are eliminated in the cell by the enzymatic action of superoxide dismutase (SOD) that form $H_2O_2$ and oxygen, while other antioxidant enzymes, such as catalase and glutathione peroxidase, intervene to eliminate $H_2O_2$. Under normal physiological conditions, the rate of reactive oxygen species (ROS) generation is balanced by the rate of elimination through endogenous and dietary antioxidants such as SOD, catalase, thioredoxin, glutathione, and vitamins. Deregulation of angiotensin II signaling induces an increase in ROS, causing hypertension and cardiovascular dysfunction (*Jin & Kang, 2024*). Several cardiovascular studies suggest that selenium (Se) and tocopherol reduce the risk of cardiovascular diseases, such as coronary heart disease (*Ju et al., 2017*); in particular, NADPH is essential in regenerating antioxidant components known as reduced glutathione. Various pathways aid in the regeneration of NADPH, with the pentose phosphate pathway being the primary one. Some studies have shown that individuals with hypertension produce more reactive oxygen species and have an impaired antioxidant defense system, which increases oxidative stress. Antioxidants inhibit oxidation reactions, minimizing damage and the production of free radicals; this antioxidant deficiency inciates the onset of cardiovascular diseases (*Wang & Kang, 2020*).

Se is an essential trace element required for optimal human health, and it is incorporated into selenocysteine and selenomethionine, which are necessary for the synthesis of selenoproteins; these participate in oxidative stress, inflammation and immunity (*Adadi et al., 2019*). Se deficiency is a serious problem in more than 500 million people in the

world (*Dinh et al., 2018*); it is associated with multiple cardiovascular diseases, including myocardial infarction, heart failure, coronary heart disease, and atherosclerosis (*Shimada, Alfulaij & Seale, 2021*). Some epidemiological studies in humans indicate a positive association between low blood Se concentration and hypertension, known as Keshan disease (*Huang et al., 2022*). The symptoms of the disease are cured with Se supplementation (*Liao et al., 2024*). The effect of Se on human health is channeled with glutathione peroxidase (GPx) activities to control blood pressure. Se is the main component of GPx; GPx activity reduces lipid peroxidation, atherosclerotic plaque formation, and platelet aggregation (*Mansour et al., 2017*). Se intake improves GPx activity and protects against hypertension and myocardial infarction (*Handy & Loscalzo, 2022*). Hypertension affects the antioxidant defense system and is partly a consequence of some dietary factors such as high sodium, fat, and refined carbohydrate content (*Lei et al., 2023*). It has been reported that Se supplementation is used in aquatic animals. (*Kohshahi et al., 2019*; *Mansour et al., 2017*) and poultry (*Khajeh Bami et al., 2022*; *Zhang et al., 2020*). Improves the growth, survival, and activity of some antioxidant enzymes. However, other studies assess the effects of supplementary Se on the growth performance and antioxidant status in goats (*Bano et al., 2019*; *Mojapelo & Lehloenya, 2019*). In this context, since meat is a food high in protein, it is reasonable to consider it a good source of bioactive peptides (*Ashaolu, Le & Suttikhana, 2023*). Diverse bioactivities such as antioxidant, antihypertensive, antithrombotic and antidiabetic effects have been reported in peptides derived from fish, bovine, chicken, pork, and duck meats (*Bezerra et al., 2019*; *Li et al., 2020*; *Maky & Zendo, 2021*; *Verma et al., 2018*).

This hypothesizes that adequate or high levels of Se in muscle tissue may enhance the antioxidant activity of myocytes and increase the angiotensin-converting enzyme (ACE) inhibitory activity. This could prevent the production of angiotensin II, a key player in the development of cardiovascular diseases. Importantly, this study is the first to investigate the ACE inhibitory activity of enzymatic hydrolysates derived from kids-goat meat supplemented with different Se sources. The findings of this study could significantly contribute to our understanding of the role of Se and ACE. Additionally, the effect of these supplements on the chemical composition of the meat and liver tissue was also evaluated, providing valuable insights into the potential health benefits of Se supplementation.

## MATERIALS & METHODS

### Animals

The study was conducted using 45 kids goat of the Pastoreña breed (average weight of $4.6 \pm 1.14$ kg, average age of 30 days) obtained from Oaxaca. The animals were transported to Colegio de Postgraduados, Montecillo campus (Texcoco, State of Mexico) under comfortable environmental conditions, specifically precisely of temperature 18.5 °C and relative humidity 60%. Kids-goats were fed exclusively goat milk and *ad libitum*. All procedures performed with animals in this study were approved by the Animal Ethics Committee of the Colegio de Postgraduados (approval reference code: 12013008) and also by the national guidelines for animal care under NOM-033-ZOO-1995.

## Experimental treatments

After a 7-day adaptation period prior to the stage before starting the experiment, the animals were randomly assigned to three groups ($n = 15$ for each experimental group) using a straightforward randomization method based on their weight as a variable for randomization. The experimental treatments were organized as follows: (1) CG: group without selenium (Se) supplementation; (2) GSS: group receiving a single subcutaneous injectable dose of sodium selenite ($Na_2SeO_3$) at a dose of 0.25 mg/kg of body weight; (3) GSM: group receiving oral administration of selenomethionine (SeMe) at a dose of 0.3 mg/kg of body weight).

## Meat samples

For each experimental treatment (*i.e.,* CG, GSS, and GSM), five animals were subjected to fasting for 14 h and moved to the COLPOS slaughterhouse, 500 m away. The euthanasia of the kids was performed with a non-penetrating captive bolt stunner (CASH® Small Animal Tool), producing immediate insensibility. It was slaughtered in appropriate conditions according to NOM-033-ZOO-1995 at 7, 14, and 21 days. From each sacrificed animal, samples of *Biceps femoris* (BF) muscles were cut from the left side of the carcasses after 4 h post-mortem. The meat samples were collected in Ziploc bags identified by treatment and kid number and frozen at $-18\,°C$ until analysis.

## Proximate composition analyses of meat

The chemical analysis of the meat was determined following the official methods of the Association of Official Analytical Chemists (AOAC) (*Horwitz & Latimer, 2005*). In this sense, the moisture (AOAC Method 934.01) was determined by weight loss after drying the sample at $100-110\,°C$ for 18 h; protein (AOAC Method 2001.11) by the micro Kjeldahl method by digesting the sample with sulfuric acid and then distillation with sodium hydroxide using a nitrogen-to-protein conversion factor value of 6.25; fat (AOAC Method 920.39) was determined by the continuous extraction method in a Goldfish system using petroleum ether as a solvent; and ashes (AOAC method 942.05) by incineration at $600\,°C$ until total loss of organic matter. All sample meat was made in duplicate. In addition, the chemical composition of the liver (*i.e.,* protein, fat, moisture, and collagen contents) was measured using near-infrared spectroscopy (FoodScan™ Lab, Foss, Sweden) according to *Anderson (2007)*.

## Preparation of enzymatic hydrolysates derived from the meat of kids-goat

Enzymatic hydrolysates from the meat of kids' goats were obtained, according to *Escudero Fernández (2010)*. First, protein concentrates were extracted by freeze-drying meat samples from kids' goats. Briefly, samples were freeze-dried for 48 h in a Labconco FreeZone 6 Freeze Dryer (Labconco Corp., Kansas City, MO, USA) at $-51\,°C$ under a vacuum pressure of 0.120 mBar. Afterward, freeze-dried samples were ground in an electric coffee grinder (Hamilton Beach®) for 30 s to get dry powders. Subsequently, dry powders were defatted with petroleum ether, and the defatted materials were stored at $-20\,°C$ until hydrolysis. Next, four grams of each dry powder sample were resuspended in 43 mL of

distilled water, and pH was adjusted to 2.0 with 0.1 M. These suspensions were heated at 37 °C and hydrolyzed for two hours with pepsin using an enzyme-to-substrate ratio of 1:100. Hydrolysis was terminated by heating to 95 °C for 10 min.

## Degree of hydrolysis

The degree of hydrolysis (DH), defined as the percent ratio of the number of peptide bonds broken (h) to the total number of bonds per unit weight (hot), was determined using the pH-stat method (*Adler-Nissen, 1986*) and calculated with the following equation:

$$DH\,(\%) = \frac{h}{h_{tot}} = \left[ \frac{\beta \times Nb}{\alpha \times M_p \times h_{tot}} \right] \times 100$$

where $\beta$ and Nb refer to the amount of NaOH used the proteolysis of the substrate and its normality, respectively; $\alpha$ represents the average degree of dissociation of the a-NH$_2$ groups in the protein substrate; M$_p$ is the mass (g) of the protein; and that denotes the total number of peptide bonds in the protein substrate. The values of h$_{tot}$ = 7.6 and $\alpha$ = 1 and the pH and temperature were regulated according to the suggestion of *Nielsen, Petersen & Dambmann (2001)*.

## Determination of protein content

The protein content of enzymatic hydrolysates derived from the meat of goat kids was estimated using the Bradford method (*Bradford, 1976*). A standard curve was constructed using bovine serum albumin (0.75, 0.5, 0.25, 0.125, 0.062 mg/mL).

## Tricine-sodium dodecyl sulfate-polyacrylamide gel electrophoresis (Tricine-SDS-PAGE)

The method was modified for gel electrophoresis. Hydrolysate samples (10 µL) were mixed with 25 µL of buffer 4x (TruPAGE 4x PCG3009, Sigma Aldrich) and 65 µL of distilled water. The resulting solution was maintained at 70 °C for 10 min, followed by cooling with ice water. A Mighty Small II SE 250 (Hoefferer, Inc.) system with TruPAGE (PCG2012-10EA; Sigma Aldrich) gels (gradient 4–20%) was utilized used. Molecular weight markers with the following molecular weights were included as a reference 23.4, 30.1, 43.6, 52.4, 63, 70.7, 79.4, 100, 123, 147.9, 162.1, 177.8, and 282 kDa. Electrophoresis was conducted at 25 mA (120 V) for 60 min. A staining solution containing Coomassie Brilliant Blue R-250 (Biorad 1610400) was applied to stain the gels for 30 min, followed by de-staining with 50% ethanol.

## Angiotensin-converting enzyme inhibitory activity

With some modifications, ACE inhibitory activity was evaluated as described by *Hayakari, Kondo & Izumi (1978)*. Samples (40 µL) were combined with 20 µL of angiotensin-converting enzyme (100 mU) and incubated for 5 min at 37 °C. Next, 100 µL of the substrate, Hippuryl-His-Leu (HHL) 0.3% (w/v), and the reaction was carried out for 45 min at 37 °C. After that, 360 µL of 2,4,6-trichloro-s-triazine dissolved in dioxane and 720 µL of potassium phosphate buffer (0.2 M, pH 8.3) were added to terminate the reaction. Later, the resulting solution was centrifuged (10,000 g, 10 min), and the absorbance of

the supernatant was recorded at 382 nm (Multiskan Go; Thermo Fisher Scientific).

$$\text{ACE inhibition}(\%) = \frac{(A-B)}{(A-C)} x 100$$

where: A denotes absorbance in the presence of ACE and sample; B is the absorbance of the control, and C refers to the absorbance of the reaction blank.

El $IC_{50}$ (ACA en kDa) before and after hydrolisis was calcul;ated with a regresion equation using the following formula: $y = mx + b$.

## Statistical analysis

The data were statistically analyzed as a complete randomized design with a factorial arrangement of 3 × 3 (samples days ×treatments), using the PRO MIXED model procedure performed on SAS software version 9.0. The factors were the experimental treatment supplements with Se (*i.e.,* CG, GSS, and GSM) and the slaughter time (*i.e.,* 7, 14, and 21 d). The terms of the model were the experimental treatment, slaughter time (repeated measures), and the interaction of both. The means of each factor were compared using the Tukey test, and differences were considered significant when $p < 0.05$. The following statistical model was used:

$$Y_{ijk} = \mu + T_i + S_j + \varepsilon_{ijk}$$

where: $Y_{ijk}$ = variable response; $\mu$= mean; Ti = effect of experimental treatment; Sj = effect of slaughter time; and $\varepsilon_{ijk}$ = random error.

## RESULTS

### Chemical composition of meat

The physicochemical composition of meat derived from kids-goat-fed diets with Se is shown in Table 1. Our results indicated no effects of the experimental treatment and time factors ($p > 0.05$) on the moisture content of the meat samples; thus, their values did not change between treatments and slaughter days. In contrast, the kids-goat meat from the GSM group exhibited a higher protein content ($p < 0.05$) than the three slaughter times and with other experimental groups. At 21 days of slaughter, the GSM treatment had approximately 5% more ($p < 0.05$) protein compared to the group without Se supplementation (CG). Conversely, the fat content gradually increased over time in the GSM treatment, which rose approximately two-fold (from 1.77 to 3.68). Similarly, the fat content increased about 1.5 times at 21 days in the GSM treatment compared to the CG treatment. In contrast, the ash content in both CG and GSS treatments decreased significantly over time ($p < 0.05$), while in the GSM treatment, there was a significant increase ($p < 0.05$) over time.

### Chemical composition of liver

The contents of moisture, protein, fat, and collagen of the liver-derived from kids-goat-fed diets with Se are shown in Table 2. For the treatments CG and GSM, the moisture content increased significantly ($p < 0.05$) over time, increasing 2.03% and 1.13%, respectively, from 7 to 21 d of slaughter. For the treatment GSS, the moisture content decreased significantly ($p < 0.05$) at 21 d compared to the values of 7 and 14 d. Similarly, the protein content of

**Table 1 Physicochemical composition of the meat (*Biceps femoris*) of kids-goat supplemented with selenium.**

| Component, % wet basis | Slaughter time | | |
|---|---|---|---|
| | 7 | 14 | 21 |
| **Moisture** | | | |
| CG | $77.00 \pm 0.30^{ax}$ | $77.09 \pm 2.09^{ax}$ | $79.20 \pm 1.15^{ax}$ |
| GSS | $78.07 \pm 0.65^{ax}$ | $76.75 \pm 0.46^{ax}$ | $72.48 \pm 4.42^{ax}$ |
| GSM | $77.42 \pm 0.02^{ax}$ | $76.41 \pm 0.52^{ax}$ | $77.24 \pm 0.48^{ax}$ |
| **Protein** | | | |
| CG | $14.95 \pm 0.44^{ax}$ | $15.57 \pm 0.23^{ax}$ | $14.26 \pm 0.10^{ax}$ |
| GSS | $15.12 \pm 0.13^{axy}$ | $16.11 \pm 0.15^{ax}$ | $15.69 \pm 0.20^{ax}$ |
| GSM | $16.49 \pm 0.35^{ay}$ | $16.78 \pm 0.12^{ax}$ | $19.43 \pm 0.28^{by}$ |
| **Fat** | | | |
| CG | $2.23 \pm 0.02^{ax}$ | $3.13 \pm 0.00^{bx}$ | $2.59 \pm 0.04^{cx}$ |
| GSS | $2.40 \pm 0.56^{ax}$ | $3.78 \pm 0.04^{bz}$ | $3.45 \pm 0.09^{cyz}$ |
| GSM | $1.77 \pm 0.003^{ay}$ | $2.49 \pm 0.02^{by}$ | $3.68 \pm 0.06^{cy}$ |
| **Ash** | | | |
| CG | $1.09 \pm 0.001^{axy}$ | $1.06 \pm 0.004^{axy}$ | $0.98 \pm 0.009^{bxy}$ |
| GSS | $1.06 \pm 0.008^{axy}$ | $1.06 \pm 0.008^{axy}$ | $1.01 \pm 0.004^{bxy}$ |
| GSM | $1.10 \pm 0.005^{axz}$ | $1.09 \pm 0.004^{axz}$ | $1.25 \pm 0.008^{bz}$ |

**Notes.**

CG, group without selenium supplementation; (2) GSS, group with a single injectable dose subcutaneously of sodium selenite ($Na_2SeO_3$) at a dose of 0.25 mg/kg of body weight; (3) GSM, group with an oral administration of selenomethionine at a dose of 0.3 mg/kg of body weight).

a–c Different superscripts among values in each row indicate significant differences ($p < 0.05$).

x–z Different superscripts among values in each column indicate significant differences ($p < 0.05$).

GSS increased significantly ($p < 0.05$) over time, while for the treatments, CG and GSM had a contrary behavior, decreasing significantly ($p < 0.05$) from 7 to 21 d of slaughter. The fat content had the same behavior as protein content; the treatment GSS with more fat (*ca.* 40%) at 21 d of slaughter. In contrast, the treatments with Se (GSS and GSM) had more collagen at 7 and 14 d of slaughter, while on the contrary, at 21 d, the CG had more collagen compared to those groups supplemented with Se.

## Degree of hydrolysis

The meat percentage of ACE inhibition before hydrolysis improved at 7d of slaughter (12.1%, $p < 0.05$) with GSM and at 14 (54%, $p < 0.05$) and 21d (49%, $p < 0.05$) of slaughter with GSS and GSM, respectively. The experimental treatment and time factors ($p < 0.05$) affected the degree of hydrolysis of the meat samples hydrolysates. It was observed that at 14 d, the degree of hydrolysis decreased ($p < 0.05$) in the treatments supplemented with Se (GSS and GSM), while at 21 d of slaughter, had a contrary behavior, with the treatments GSS and GSM having major ($p < 0.05$) degree of hydrolysis compared to the group without Se supplementation (CG). In particular, it was observed that the degree of hydrolysis increased ($p < 0.05$) significantly over time in those treatments with Se (GSS and GSM).

**Table 2  Physicochemical composition of the liver of kids-goat supplemented with selenium.**

| Component (% wet basis) | Slaughter time | | |
|---|---|---|---|
| | 7 | 14 | 21 |
| Moisture | | | |
| CG | 64.39 ± 0.14$^{ax}$ | 65.71 ± 0.11$^{bx}$ | 66.42 ± 0.07$^{cx}$ |
| GSS | 66.98 ± 0.05$^{az}$ | 66.53 ± 0.13$^{abz}$ | 65.97 ± 0.009$^{bx}$ |
| GSM | 65.81 ± 0.10$^{ay}$ | 67.20 ± 0.11$^{by}$ | 66.94 ± 0.04$^{bx}$ |
| Protein | | | |
| CG | 24.25 ± 0.14$^{ax}$ | 22.91 ± 0.08$^{bx}$ | 19.69 ± 0.013$^{cx}$ |
| GSS | 20.51 ± 0.01$^{az}$ | 20.80 ± 0.04$^{az}$ | 21.27 ± 0.17$^{bz}$ |
| GSM | 22.08 ± 0.03$^{ay}$ | 21.59 ± 0.01$^{by}$ | 20.26 ± 0.03$^{cy}$ |
| Fat | | | |
| CG | 4.38 ± 0.02$^{ax}$ | 4.61 ± 0.05$^{bx}$ | 2.71 ± 0.03$^{cx}$ |
| GSS | 2.86 ± 0.01$^{az}$ | 3.63 ± 0.03$^{bz}$ | 4.05 ± 0.02$^{cz}$ |
| GSM | 4.03 ± 0.05$^{axy}$ | 2.66 ± 0.02$^{by}$ | 3.25 ± 0.04$^{cy}$ |
| Collagen | | | |
| CG | 1.52 ± 0.09$^{x}$ | 1.47 ± 0.02 | 1.94 ± 0.18 |
| GSS | 2.04 ± 0.13$^{y}$ | 1.91 ± 0.11 | 1.71 ± 0.02 |
| GSM | 1.90 ± 0.06$^{xy}$ | 1.90 ± 0.05 | 1.86 ± 0.02 |

Notes.
Experimental treatment codes are defined in Table 1.
a–c Different superscripts among values in each row indicate significant differences ($p < 0.05$).
x–z Different superscripts among values in each column indicate significant differences ($p < 0.05$).

## Protein content

Overall, both before and after hydrolysis, the protein content significantly decreased ($p < 0.05$) over time in all treatments (5). Similarly, before and after hydrolysis at 21 days post-slaughter, the treatments supplemented with Se (GSS and GSM) exhibited higher ($p < 0.05$) protein content compared to the group without Se supplementation (CG). In general, the protein content in samples prior to hydrolysis was greater than after hydrolysis (Table 3).

## Tricine -SDS-PAGE electrophoresis

The electrophoretic patterns of the enzymatic hydrolysates at zero hours and two hours of hydrolysis obtained from the enzymatic hydrolysates derived from the meat of kid goats supplemented with Se. It can be observed that the native protein patterns at zero hours of hydrolysis of meat from kid goats supplemented with Se were very similar across all treatments. These non-hydrolyzed proteins exhibited a molecular weight between 37.1 and 213.7 kDa. In contrast, the electrophoretic patterns of the enzymatic hydrolysates at two hours showed a molecular weight range between 23.44 and 27.5 kDa, the bands exhibiting greater intensity. At 21 days of post-slaughter, a higher degree of hydrolysis was observed in the treatments supplemented with Se (GSS and GSM) compared to the group without Se supplementation (CG) (Fig. 1). Notably, it was observed that the degree of hydrolysis increased ($p < 0.05$) significantly over time in the treatments with Se (GSS and GSM).

**Table 3** Protein content, percentage of inhibition of ACE and $IC_{50}$ values determined with the molecular weight (KDa) of the angiotensin-converting enzyme inhibitory activity in goat meat supplemented with two sources of selenium.

| | Slaughter time | | |
|---|---|---|---|
| | **7** | **14** | **21** |
| **Protein content** | | | |
| Before hydrolysis | | | |
| CG | $2.23 \pm 0.02^{ax}$ | $1.88 \pm 0.008^{bx}$ | $1.55 \pm 0.02^{cx}$ |
| lGSS | $1.88 \pm 0.016^{az}$ | $1.84 \pm 0.02^{ax}$ | $1.84 \pm 0.02^{ayz}$ |
| GSM | $2.59 \pm 0.043^{ay}$ | $1.94 \pm 0.05^{bx}$ | $1.83 \pm 0.01^{bcy}$ |
| After hydrolysis | | | |
| CG | $0.13 \pm 0.002^{ax}$ | $0.12 \pm 0.001^{bx}$ | $0.09 \pm 0.00^{cx}$ |
| GSS | $0.13 \pm 0.002^{az}$ | $0.10 \pm 0.00^{bz}$ | $0.11 \pm 0.00^{cz}$ |
| GSM | $0.16 \pm 0.001^{by}$ | $0.12 \pm 0.001^{by}$ | $0.10 \pm 0.001^{cy}$ |
| **Percentage of inhibition of ACE** | | | |
| Before hydrolysis | | | |
| CG | $22.80 \pm 1.03^{ax}$ | $9.17 \pm 0.61^{bx}$ | $16.19 \pm 1.26^{cx}$ |
| GSS | $21.37 \pm 0.36^{ax}$ | $14.88 \pm 1.11^{bz}$ | $19.99 \pm 1.89^{aby}$ |
| GSM | $25.55 \pm 0.90^{ay}$ | $23.36 \pm 0.94^{by}$ | $24.18 \pm 1.52^{abz}$ |
| After hydrolysis | | | |
| CG | $90.54 \pm 0.03^{ax}$ | $92.84 \pm 0.06^{bx}$ | $93.57 \pm 0.06^{cx}$ |
| GSS | $91.51 \pm 0.08^{abz}$ | $92.69 \pm 0.06^{bx}$ | $94.38 \pm 0.05^{cy}$ |
| GSM | $92.40 \pm 0.01^{byz}$ | $91.79 \pm 0.06^{by}$ | $93.76 \pm 0.08^{cx}$ |
| **$IC_{50}$ values determined with the molecular weight (KDa)** | | | |
| Before hydrolysis | | | |
| CG | $0.565 \pm 0.03^{ax}$ | $0.624 \pm 0.61^{bx}$ | $0.627 \pm 0.26^{bx}$ |
| GSS | $0.627 \pm 0.36^{ay}$ | $0.626 \pm 0.11^{ax}$ | $0.623 \pm 0.89^{axy}$ |
| GSM | $0.631 \pm 0.90^{ay}$ | $0.624 \pm 0.94^{bx}$ | $0.618 \pm 0.52^{aby}$ |
| After hydrolysis | | | |
| CG | $0.726 \pm 0.04^{ax}$ | $0.676 \pm 0.5^{bx}$ | $0.715 \pm 0.05^{cx}$ |
| GSS | $0.679 \pm 0.07^{aby}$ | $0.715 \pm 0.4^{bcy}$ | $0.685 \pm 0.04^{acx}$ |

**Notes.**

Experimental treatment codes are defined in Table 1. Before hydrolysis was at 0 h of hydrolysis and after hydrolysis was at 2 h of hydrolysis.

a–c Different superscripts among values in each row indicate significant differences ($p < 0.05$).

x–z Different superscripts among values in each column indicate significant differences ($p < 0.05$).

## ACE inhibitory activity of enzymatic hydrolysates derived from meat samples

Overall, the ACE inhibitory activity of samples after hydrolysis was higher than that of samples before hydrolysis (Table 3). ACE inhibitory activity values after hydrolysis were 4–6 times greater than those before hydrolysis. Following hydrolysis, the ACE inhibition percentage for all treatments significantly increased ($p < 0.05$) over time ($p < 0.05$). Notably, the meat protein content and rate of ACE inhibition after hydrolysis improved (50% and 2%, $p < 0.05$) with GSM at 7 days post-slaughter. The $IC_{50}$ values are determined by the molecular weight of ACE before and after hydrolysis. After hydrolysis, the $IC_{50}$ of

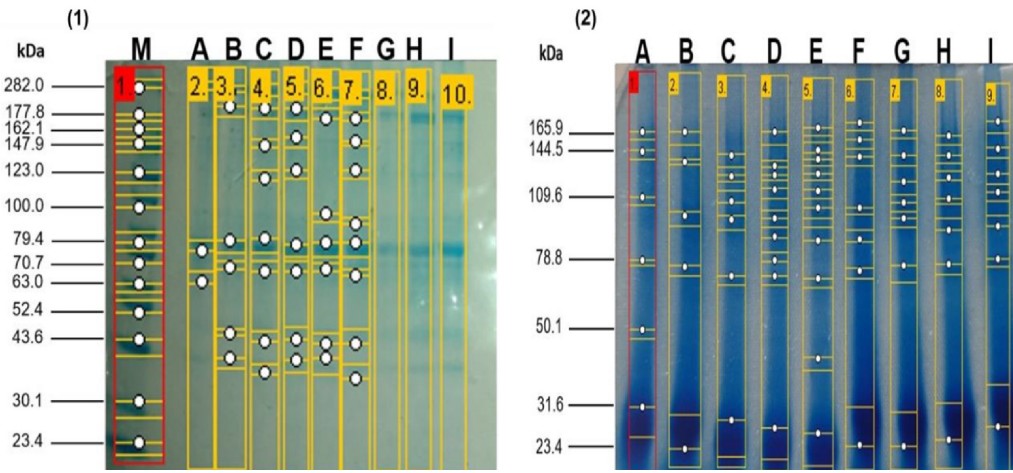

**Figure 1** SDS-PAGE profile from meat proteins of kids-goat supplemented with different selenium sources before (1) and after (2) enzymatic hydrolysis. M: Molecular weight markers; A, D G: Treatments without Se supplementation at 7, 14, and 21 d of slaught.

the selenium-supplemented groups decreased ($p < 0.05$), along with the amount of CAE and $IC_{50}$ values (Table 3).

# DISCUSSION

## Chemical analysis of meat and liver

The physicochemical composition of meat derived from kid-goat-fed diets with Se showed no effects of the experimental treatment and time factors on moisture content. Thus, Se supplementation did not influence the moisture content of the meat samples, which were between 72.5–79.2%. Similar observations in moisture contents were reported in Indian kids-goat meat with values of 72.4–75.3% (*Das & Rajkumar, 2010*), Brazilian kids-goat (75.9–76.9%) (*Freitas et al., 2011*), and kids-goat originating from East Africa (70.7%) (*Shija et al., 2013*). In a related study, the supplementation of organic Se (Se yeast) at the dose of 0.3 mg/kg diet showed 51.8% moisture in 4-month kids-goat, which was below compared to the data of our study. The protein content in meat samples was increased by the supplementation of Se, which could be a result of the hypertrophic effect of Se in the muscles of goats (*Parveen Samo et al., 2018*; *Rannem et al., 1995*). In addition, it has been reported that supplementation with minerals such as Se in the diet influences growth and nutrient digestibility rates (*Parveen Samo et al., 2018*). For example, *Del Razo-Rodriguez et al. (2013)* reported that the total tract digestibility of lambs was increased because of Se supplementation at a dose of 0.2−0.6 mg/kg. On the other hand, kids-goat meat contains a low-fat content clow fat to other meats such as beef or pork (>10% fat) (*Park et al., 2018*). In the literature on goat meat, the fat content was reported between 2.2−4.05% (*De Palo et al., 2015*; *Ivanović et al., 2020*), consistent with our findings. Our data suggest that Se supplementation increased the fat *ca.* 43% at 21 d of slaughter compared to 7 d of slaughter. In a related study, similar results were obtained

by *Parveen Samo et al. (2018)*, showing a 37% fat in the group supplemented with Se compared to the group without it. Se supplementation has been reported to alter lipid metabolism by decreasing cholesterol deposition in muscle, possibly by changing the ratio between reduced glutathione and oxidized glutathione (*Netto et al., 2014*). This may suggest meat with desirable characteristics. For example, it has been reported that the meat's taste, tenderness, and succulence of the meat are influenced by collagen content and other factors like pH, sarcomere length, streaky fat, and the degree of muscle protein degradation (*Erasmus, Muller & Hoffman, 2017*; *Martínez Marín, 2008*).

As mentioned above, our findings suggest that supplementation improves certain chemical components of goat meat from kids, providing desirable quality attributes for commercialization. To the best of our knowledge, this is the first study reporting the effect of selenium (Se) diets on the chemical composition of liver derived from kids'goats, which can serve as reference values for future research. Overall, there was no general effect of Se on the liver's chemical composition over time. However, it was noted that the experimental treatment GSS, a group of kids' goats supplemented with Se through a subcutaneous injectable dose, increased their protein, fat, and collagen content compared to the group without Se supplementation (CG) or the group administered Se orally. This indicates that the route of Se administration can influence its effect on liver composition. For instance, organic Se is considered more bioavailable than inorganic forms (*Amoako, Uden & Tyson, 2009*). However, further studies are needed to validate this observation.

### Degree of hydrolysis, protein content, and SDS-PAGE analysis

In our study, the degree of hydrolysis was influenced by the Se supplementation and the time of slaughter. This parameter indicates the efficiency of the hydrolysis of meat proteins to produce bioactive peptides (*Bahari et al., 2020*), so a higher degree of hydrolysis may imply an increase in both the number and size of peptides, which can enhance their bioactivities (*Upata et al., 2022*). Therefore, selenium positively affects the degree of hydrolysis of meat, although no prior studies have been found on the impact of selenium on the hydrolysis of animal products. Concurrently, there is evidence of increased activity of certain hydrolytic enzymes in plants due to selenium's influence (*Zeid et al., 2019*); a similar or alternative mechanism could occur in meat. In contrast, the protein content decreased in all treatments after the hydrolysis, consistent with previous reports on fish (*Wisuthiphaet & Kongruang, 2015*) and chicken (*Yuliatmo, Fitriyanto & Bachruddin, 2017*) protein hydrolysates. According to these authors, larger proteins were converted into smaller peptides and amino acids after hydrolisis, while the methods used to quantify proteins in hydrolysates (*i.e.,* Bradford, Lowry) can only detect proteins with a molecular weight greater than 3,000 Da. SDS-PAGE confirmed both observations regarding the degree of hydrolysis and protein content following hydrolysis, larger proteins were degraded into smaller peptides and amino acids. Based on the results, a greater degree of hydrolysis may indicate that the meat proteins were further broken down into multiple polypeptide chains, small peptides, and amino acids. Similar results on SDS-PAGE were reported in hydrolysates of Korean native cattle (*Lee & Hur, 2017*) and spent hen meat (*Kumar et al., 2021*).

## ACE inhibitory activity

Angiotensin-I converting enzyme (ACE) is related to controlling blood pressure and electrolyte homeostasis by converting angiotensin I into potent vasoconstrictor angiotensin II (*Ahmad et al., 2023*). Thus, the inhibition of ACE can help to regulate the blood pressure in hypertensive persons and the treatment of various cardiovascular diseases such as heart failure, myocardial infarction, diabetic nephropathy, or renal dysfunction (*Ktari et al., 2014*). Although current medical treatments are very effective (*e.g.*, captopril, enalapril, alacepril, or lisinopril), they showed several side effects (inflammatory response, dry cough, taste disturbance, or angioneurotic edema in some patients (*Ambigaipalan, Al-Khalifa & Shahidi, 2015*). Our data indicated that ACE inhibitory activity increased five-fold over time in all treatments, particularly in groups with Se supplementation. All hydrolysates demonstrated ACE inhibitory activity greater than >90%, more significant. This study shows that kids-goat meat hydrolysates can inhibit ACE activity. The ACE inhibitory activity of these peptides may be attributed to their distinct amino acid compositions and hydrophobicity. Scientific literature suggests that hydrophobic amino acid residues such as leucine, valine, alanine, tryptophan, tyrosine, proline, and phenylalanine preferentially bind to the catalytic sites of ACE. Therefore, these peptides can act as strong competitive ACE inhibitors (*Jung et al., 2006*; *Wu, Liao & Udenigwe, 2017*). However, further studies are necessary to characterize the peptides responsible for the observed bioactivities. In particular, peptidomics and mechanistic studies are needed to observe the enzyme–peptide interactions such as ACE-peptides.

## CONCLUSIONS

This is the first report describing the ACE inhibitory activity of bioactive peptides derived from Se-supplemented goat meat. Selenomethionine dosing for 21 days improved the total protein content of the meat by 36%. On the other hand, the percentage of ACE inhibition in the meat before hydrolysis was enhanced at 7 days of slaughter with selenomethionine and 14 and 21 days of slaughter with the application of both selenium sources, respectively. These results indicate the presence of ACE in goat meat; however, the percentage of ACE inhibition after hydrolysis was only improved with selenomethionine dosing at 7 days of slaughter. The lack of ACE recovery in the other two slaughter periods was likely due to a lack of precision in the purification technique. The study's novelty indicates that supplemented selenium had a synergism with ACE in goat meat. It is necessary to continue these studies to identify specific bioactive peptides, antioxidant activities, and goat meat's biological and functional value, considering it a functional food that can prevent metabolic diseases and be a healthy alternative for the human population.

### Funding

This study was supported by the Universidad Autonoma Metropolitana, Colegio de Postgraduados Mexico, and by the Scholarship of Consejo Nacional de Humanidades

Ciencia y Tecnologi'a (CONAHCyT), Me'xico. The funders had no role in study design, data collection and analysis, decision to publish, or preparation of the manuscript.

### Grant Disclosures

The following grant information was disclosed by the authors:
The Universidad Autonoma Metropolitana, Colegio de Postgraduados Mexico.
The Scholarship of Consejo Nacional de Humanidades Ciencia y Tecnologi'a (CONAHCyT), Me'xico.

### Competing Interests

The authors declare there are no competing interests.

### Author Contributions

- Silvia C. Pérez-Ramirez conceived and designed the experiments, prepared figures and/or tables, and approved the final draft.
- Rosy Cruz-Monterrosa conceived and designed the experiments, prepared figures and/or tables, and approved the final draft.
- Mayra Diaz-Ramirez performed the experiments, authored or reviewed drafts of the article, and approved the final draft.
- Erika B. León-Espinosa analyzed the data, authored or reviewed drafts of the article, and approved the final draft.
- José E. Aguilar-Toalá analyzed the data, authored or reviewed drafts of the article, and approved the final draft.
- Monzerrat Rosas-Espejel analyzed the data, prepared figures and/or tables, and approved the final draft.
- J. Efren Ramirez-Bribiesca conceived and designed the experiments, performed the experiments, authored or reviewed drafts of the article, and approved the final draft.

### Animal Ethics

The following information was supplied relating to ethical approvals (i.e., approving body and any reference numbers):

Colegio de Postgraduados provided approval for all procedures performed with animals in this study (approval reference code: 12013008).

### Field Study Permissions

The following information was supplied relating to field study approvals (i.e., approving body and any reference numbers):

Colegio de Postgraduados, Mexico, approved his study (12013008).

### Data Availability

The raw data is available in the Supplemental File.

### Supplemental Information

Supplemental information for this article can be found online at http://dx.doi.org/10.7717/peerj.19261#supplemental-information.

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
