# Peer review of "Chemical analysis and angiotensin I-converting enzyme inhibitory activity of enzymatic hydrolysates derived from meat of goat-kids with supplemental selenium"

_PeerJ, doi:10.7717/peerj.19261_

## Round 0.1 · original submission · Major Revisions

Please look at the reviewers' comments and requests and submit a thoroughly revised version with a detailed rebuttal letter.

**Language Note:** The review process has identified that the English language must be improved. PeerJ can provide language editing services - please contact us at [email protected] for pricing (be sure to provide your manuscript number and title). Alternatively, you should make your own arrangements to improve the language quality and provide details in your response letter. – PeerJ Staff

Reviewer 1 ·

Basic reporting

no comment

Experimental design

no comment

Validity of the findings

no comment

Additional comments

Regarding the manuscript entitled “Chemical analysis and Angiotensin I-converting enzyme inhibitory activity of enzymatic hydrolysates derived from meat of goat-kids with supplemental selenium. This article is suitable for published since it contains much valuable information but after the authors do major modification.

1. The fraction of Introduction suggested to be reworded to highlight the novelty.
2. The aim of this manuscript is not clear.
3. The pH, temperature and duration of hydrolysis that was conducted – what was the basis for these? Any optimization work done to determine these parameters?
4. Figure 1. It should replace with high resolution one.
5. In protein estimation of hydrolysate, how authors subtracted the hydrolyzing enzymes proteins from their hydrolysate?
6. Conclusion is always a hard part of the work to write. It’s difficult not just repeat the results and write the conclusions as a resume of the results. But it's always better to make the conclusions a space to draw attention to the relevance of the work and, inclusively, open up possibilities for future studies. What was the contribution that this work brought to the knowledge about these hydrolysate? Why is it interesting to do this work? What are the next steps the team hopes to take to continue this research? I am sure that the authors can improve this conclusion.

Reviewer 2 ·

Basic reporting

In this manuscript, the authors determined the ACE inhibitory activity of bioactive peptides derived from the meat of goat-kids with supplemented with selenium. However, there are several major problems that should be addressed:

- Ensure your manuscript is carefully proof read by a fluent English speaking scientist before submission. There are several spelling errors throughout the manuscript.

- Authors lack to present the background on how the supplementation of Se may benefit on the generation of bioactive peptides; since authors release peptides with enzymatic hydrolysis.

- Ensure the paragraphs in the Discussion section are not as long as presented. Please see Line 260 to 307; 310 to 328 and 332 to 351.

- After thorough analysis of the manuscript, there are no significant differences in the supplementation of Se on kids-goat meat. In fact, after enzymatic hydrolysis protein content decreased. Additionally, ACE inhibition were similar between all samples, although was higher in GSS sample (21 days). Nevertheless, we suggest that IC50 (a measure of the potency of a substance (in this case peptide content) in inhibiting a specific biological or biochemical function) may provide more insight and further discussion of the present results.

Experimental design

- Authors presented literature that Se may improve growth and antioxidant activity of goats, there is a lack of a well-defined research question. In this sense, peptides were obtained with enzymatic hydrolysis not through Se supplementation. Therefore, indicate how Se may benefit the release of peptides after enzymatic hydrolysis.

- Line 97 to 105: although you indicate that 5 animals were slaughtered every 7 days to obtain samples, raw data only shows 4 analysis.

- Authors did not indicated if ACE inhibition were determined on the whole hydrolyzed sample or did authors obtained smaller fractions.

- For hydrolysis degree OPA method is a well reported method.

- Line 162 to 175: why did you only determined inhibition (%) and not IC50?

Validity of the findings

After reviewing this manuscript, authors concluded the Se supplementation to the production of ACE inhibitory peptides. However, the present results lacked of significant differences in all results. Therefore, authors are overestimating the results and conclusion.

As suggestion, include IC50 analysis for a more robust study. Also, a correlation between Se supplementation, protein content and ACE inhibition may also provide interesting insights.

Additional comments

No comments.

---

## Round 0.2 · Minor Revisions

Please review the corrections suggested in the attached PDF file, and correct the IC50 while italicizing the "p" in probabilities.

Reviewer 1 ·

Basic reporting

no comment

Experimental design

no comment

Validity of the findings

no comment

Additional comments

no comment

---

## Round 0.3 · accepted · Accept

Thanks for making the final corrections.